# Fossil bone histology reveals ancient origins for rapid juvenile growth in tetrapods

Megan R. Whitney [1✉], Benjamin K. A. Otoo[2,3], Kenneth D. Angielczyk[2,3] & Stephanie E. Pierce[1✉]

Patterns of growth throughout the lifetime of an animal reflect critical life history traits such as reproductive timing, physiology, and ecological interactions. The ancestral growth pattern for tetrapods has traditionally been described as slow-to-moderately paced, akin to modern amphibians, with fast growth and high metabolic rates considered a specialized physiological trait of amniotes. Here, we present bone histology from an ontogenetic series of the Early Carboniferous stem tetrapod *Whatcheeria deltae*, and document evidence of fibrolamellar bone—primary bone tissue associated with fast growth. Our data indicate that *Whatcheeria* juveniles grew rapidly and reached skeletal maturity quickly, allowing them to occupy a large-bodied predator niche in their paleoenvironment. This life history strategy contrasts with those described for other stem tetrapods and indicates that a diversity of growth patterns existed at the origins of tetrapod diversification. Importantly, *Whatcheeria* marks an unexpectedly early occurrence of fibrolamellar bone in Tetrapoda, both temporally and phylogenetically. These findings reveal that elevated juvenile growth is not limited to amniotes, but has a deep history in the tetrapod clade and may have played a previously unrecognized role in the tetrapod invasion of land.

[1] Museum of Comparative Zoology and Department of Organismic and Evolutionary Biology, Harvard University, Cambridge, MA 02138, USA. [2] Committee on Evolutionary Biology, University of Chicago, Chicago, IL 60637, USA. [3] Negaunee Integrative Research Center, Field Museum of Natural History, Chicago, IL 60605-2496, USA. ✉email: mwhitney@luc.edu; spierce@oeb.harvard.edu

The rate at which an animal grows is a critical component of its life history and ecology. As such, the relationships between growth rates and reproductive timing, fecundity, body size, trophic interactions, resource availability, and physiology have been of considerable interest[1–5]. The evolution of growth rates is commonly discussed in the context of a life history trade-off, whereby reduced growth rates delay reproductive timing and fast growth rates incur a heavy burden on energy allocation[5]. Among modern terrestrial vertebrates, birds and mammals are known to have the highest rates of growth, facilitated by sustained elevated rates of metabolic activity[3,6]. Given that elevated growth was once thought to be restricted to crown mammals and birds, these physiological traits were long considered a derived adaptation that arose well within amniotes[3,6–8].

The application of bone histology to the fossil record has expanded the phylogenetic scope of elevated growth rates in vertebrates. Bone histology acts as a key proxy for somatic growth rates as these data are captured in the organization of bone tissue deposition during life[9–12]. Bone tissue that is composed of both haphazardly arranged matrix fibers and vascular spaces is correlated with high somatic growth at the time of bone deposition. The bone tissue type most tightly correlated with high growth rates is fibrolamellar bone—a composite tissue that contains both a periosteal base of woven, disorganized matrix and abundant vascular spaces that are supported by concentrically deposited lamellar and/or parallel-fibered bone[9,10,13]. Fibrolamellar bone is the primary tissue type of modern bird and mammal bone but it also occurred in a range of fossil taxa including dinosaurs[14], pterosaurs[15], archosauromorphs[16], and non-mammalian synapsids including early 'pelycosaurs'[17–19]. These fossil data have pushed back the origins of rapid growth, implying that at least some of the first amniotes (~320 mya) were capable of the facultatively elevated metabolisms required to produce such bone tissue[20].

The elevated growth rates observed in some stem amniotes stand in contrast to the general trends recorded in fossil and modern lissamphibians (although there are exceptions e.g. refs. [21–23]), as well as tetrapodomorph fishes (e.g. finned *Eusthenopteron*) and stem tetrapods (e.g. limbed *Acanthostega*) that tend to display slow growth rates even early in ontogeny[24–33]. This raises the question of whether rapid growth, especially early in ontogeny, is a life history feature restricted to amniotes. However, testing this hypothesis requires histological sampling of comprehensive ontogenetic series: as bone is a dynamic tissue, it often remodels or overprints the earliest records of growth[34–36]. Consequently, a lack of detailed ontogenetic information for fossil species can lead to misinterpretations of the growth record captured in bone histology[37].

Here, we utilize the exceptional preservation of a robust ontogenetic series from *Whatcheeria deltae*, an Early Carboniferous (Mississippian) species often recovered as a member of one of the earliest-diverging post-Devonian stem tetrapod lineages[38–42] (Fig. 1). Recent anatomical investigations have revealed that *Whatcheeria* is a highly unusual animal with cranial mechanosensory lines as well as large, robust limbs and a short tail (Fig. 1), blurring the ostensible divide between aquatic and terrestrial adaptations[43–45]. This body plan is also present in the Mississippian stem tetrapod *Pederpes*[44,46], and reinterpretations of 'whatcheeriid' material suggest that rather than an anatomically plesiomorphic grade, the whatcheeriids were a Carboniferous clade with a distinct body plan.

Using both paleohistological and μCT techniques, we examined the bone histology of nine *Whatcheeria* femora that span the range of known size classes established by Otoo et al.[44] (Fig. 1). We find evidence of fibrolamellar bone in the smallest size classes that becomes completely remodeled during growth. As a result,

the femora of skeletally mature *Whatcheeria* have more 'standard' stem tetrapod bone histology, with narrow cortical walls of lamellar tissue surrounding large medullary spaces filled with a trabecular network[24,47–49]. These findings suggest that at least some stem tetrapods were capable of unexpectedly rapid growth rates early in their ontogeny, a strategy that may have reduced time to sexual and skeletal maturity while also providing a selective advantage in a time of unpredictable global and local environmental change. Therefore, instead of rapid early growth arising late in crown tetrapods, this life history strategy may have been deployed throughout tetrapod evolutionary history, including during the clade's earliest diversification.

## Results and discussion

**FMNH PR 5022.** In our smallest specimens, from size class I, the cortical walls are composed of fibrolamellar bone with reticular primary canals (Fig. 2a–h). This bone matrix becomes more organized towards the periosteum where there are increasing amounts of parallel-fibered bone and longitudinally oriented vascular canals (Fig. 2c, g). There is some evidence of secondary remodeling in the cortex (Fig. 2h) and substantial evidence of medullary remodeling (Fig. 2d). Along the endosteal surface, active erosion of the primary fibrolamellar bone has occurred as revealed by the coupling of pitted margins indicating active erosion as well as newly deposited lamellar buttressing (Fig. 2d). Remnants of the primary bone tissue are found within the medullary trabeculae suggesting that at least some of the trabecular structure formed from the scaffolding of the primary cortex (Fig. 2d). The presence of a thick cortical wall with well-vascularized tissue is consistent with an additional specimen from size class I, FMNH PR 1735, that was μCT scanned (Fig. 2e–h).

**FMNH PR 5021.** A narrow cortex composed mostly of fibrolamellar and parallel-fibered bone is present in the larger specimen from size class II (Fig. 2i–l). Although there is fibrolamellar bone with longitudinally oriented canals (Fig. 2k), it lacks the reticular fibrolamellar bone found in the size class I specimen FMNH PR 5022. There is evidence of active erosion of the cortex as both endosteal remodeling and increasing amounts of secondary remodeling (Fig. 2j). The trabeculae are comparatively more spindle-like than in FMNH PR 5022, with greater medullary space between spicules (Fig. 2i). Remnants of woven bone are present in many areas of the trabecular network, serving as a scaffold for lamellar trabecular formation (Fig. 2l).

**FMNH PR 1962.** The histology of size class III reveals a narrow cortex and a dense trabecular network in the medullary cavity (Fig. 3a). The more endosteal region of the cortex is largely remodeled and reworked and consists of parallel-fibered primary tissue with dense vasculature (Fig. 3b, c). The periosteal region of the cortex is also composed of parallel-to-lamellar-fibered bone but is largely avascular, with few periosteally-open vascular spaces (Fig. 3b–d). This arrangement suggests that outward deposition of bone tissue was either occurring slowly or had entirely ceased. There is evidence of secondary remodeling in the periosteal portion of the cortex (Fig. 3d). The narrow, poorly vascularized cortical tissues described from the histology of FMNH PR 1962 are consistent with additional specimens from size class III, FMNH PR 1952, PR 1992, PR 1760, that were μCT scanned (Fig. 3e–h).

**FMNH PR 5023.** The femora of the largest specimen from size class IV is primarily composed of a dense and highly interconnected trabecular network (Fig. 4a). The cortex that does exist is formed by lamellar bone (Fig. 4b, c, f) that is occasionally

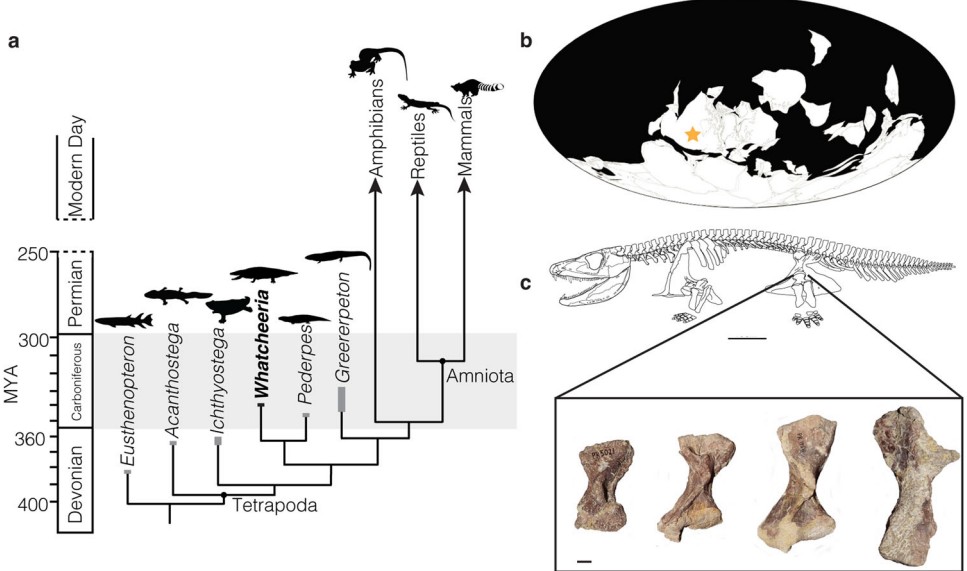

**Fig. 1 *Whatcheeria deltae* is an early diverging Carboniferous stem tetrapod. a** Known from the latest Viséan–earliest Serpukhovian (331–326 Ma) Jasper Hiemstra Quarry of Iowa, USA (**b**). Our sample includes a range of known size classes of *Whatcheeria* femora, a selection of which are figured here (**c**). From left to right the specimens include size class I (FMNH PR 5021), size class II (FMNH PR 1962), size class III (FMNH PR 1958), and size class IV (FMNH PR 5023). Scale bar for skeletal reconstruction = 10 cm and femora specimens = 1 cm. Global map is modified from the PaleoBio Database and *Ichthyostega* silhouette was drawn by SEP, while others were sourced from PhyloPic.

punctuated by vascular zones where osteonal infillings develop longitudinally oriented canals (Fig. 4b, c). There are occasional secondary osteons within the lamellar matrix, though they are smaller than the more endosteally located osteons eroding the cortical matrix (Fig. 4c). The trabeculae in this size class are well-developed and formed largely by lamellar struts and occasional secondary osteons (Fig. 4d). This organization of well-developed trabecular bone and a narrow cortex is consistent with an additional specimen from size class IV, FMNH PR 1958, that was μCT scanned (Fig. 4e, f).

**Comparative bone histology**. Within the size classes represented by our sample, we identify two critical changes in *Whatcheeria* femoral histology. As size increases there is (1) reduced cortical thickness and (2) reduced rates of bone deposition. Cortical bone in the smallest specimen from size class I (FMNH PR 5022) makes up a majority of the overall cross-sectional area (57%), contrasting with the second smallest specimen from size class II (FMNH PR 5021) where only 25% of the cross-sectional area is cortical bone. The two larger specimens from size classes III and IV have slightly larger cortical contributions, but still lower proportions than in the smallest specimen (FMNH PR 1962: 34%; FMNH PR 5023: 35%). In increasing size order, the cortices described here also vary in tissue composition with FMNH PR 5022 containing mostly fibrolamellar bone and a narrow amount of parallel-fibered bone at the outer, periosteal surface; FMNH PR 5021 containing largely parallel-fibered bone; FMNH PR 1962 containing parallel-fibered bone and lamellar bone along the periosteal surface; and FMNH PR 5023 containing mostly lamellar bone. These observations are confirmed by virtual thin sections of additional specimens as well as a more transitional specimen (FMNH PR 1952) that contains features that fall between size class II and size class III (Fig. 3e, f). Finally, no growth marks were apparent in any of the tissues examined. Even in the highly organized lamellar tissues preserved in the two largest specimens, there were no indicators of complete or partial cessations of growth. This indicates that all the individuals in our sample deposited bone without major periods of inactive growth,

although the largest specimen does show reduced bone deposition rates at its periosteal surface.

**Rapid growth rates**. The reticular fibrolamellar bone found in our histological sample of *Whatcheeria* (Fig. 2b) represents the earliest instance of this tissue type in both a temporal (i.e. Mississippian) and phylogenetic context. Fibrolamellar bone is generally deposited during the rapid juvenile growth phase of vertebrates with elevated metabolic rates[9,50–53]. It is a composite tissue type that has both a woven matrix organization that reflects the rapid depositional rate, as well as lamellar osteonal infilling in the abundant embedded vasculature spaces that provides a mechanical advantage when bones experience longitudinal loading[51,54]. As a result, fibrolamellar bone uniquely facilitates rapid size increase of a bony element while also complying with the structural demands imposed by a growing body interacting with its environment (e.g. limb loading during locomotion). The discovery of elevated growth in a deeply-rooted stem tetrapod challenges the hypothesis that such growth dynamics evolved much later and are restricted to amniotes and their close relatives[2,3,5–8]. It also raises uncertainty about the use of slow-growing modern and fossil amphibians as models for the life histories of the earliest tetrapods[24,55]. Instead, the data presented here demonstrate that fibrolamellar bone, and the rapid growth associated with this tissue type, evolved close to the origin of tetrapods.

**Growth through ontogeny**. A recent description of *Whatcheeria* postcranial material identified four size classes based on the morphology and degree of ossification of limb bones and girdles[44]. Our histological sample reveals that bone tissue organization and cortical thickness change with size class as expected from an ontogenetic growth series of a single species (Fig. 5). When examined in order of size class, there is an overall shift from fibrolamellar to parallel-fibered to lamellar bone, and a considerable thinning of the cortex, indicating a shift from rapid growth rates as expected in juveniles to the slowed growth of adults reaching skeletal maturity[53,56]. Size class I *Whatcheeria*

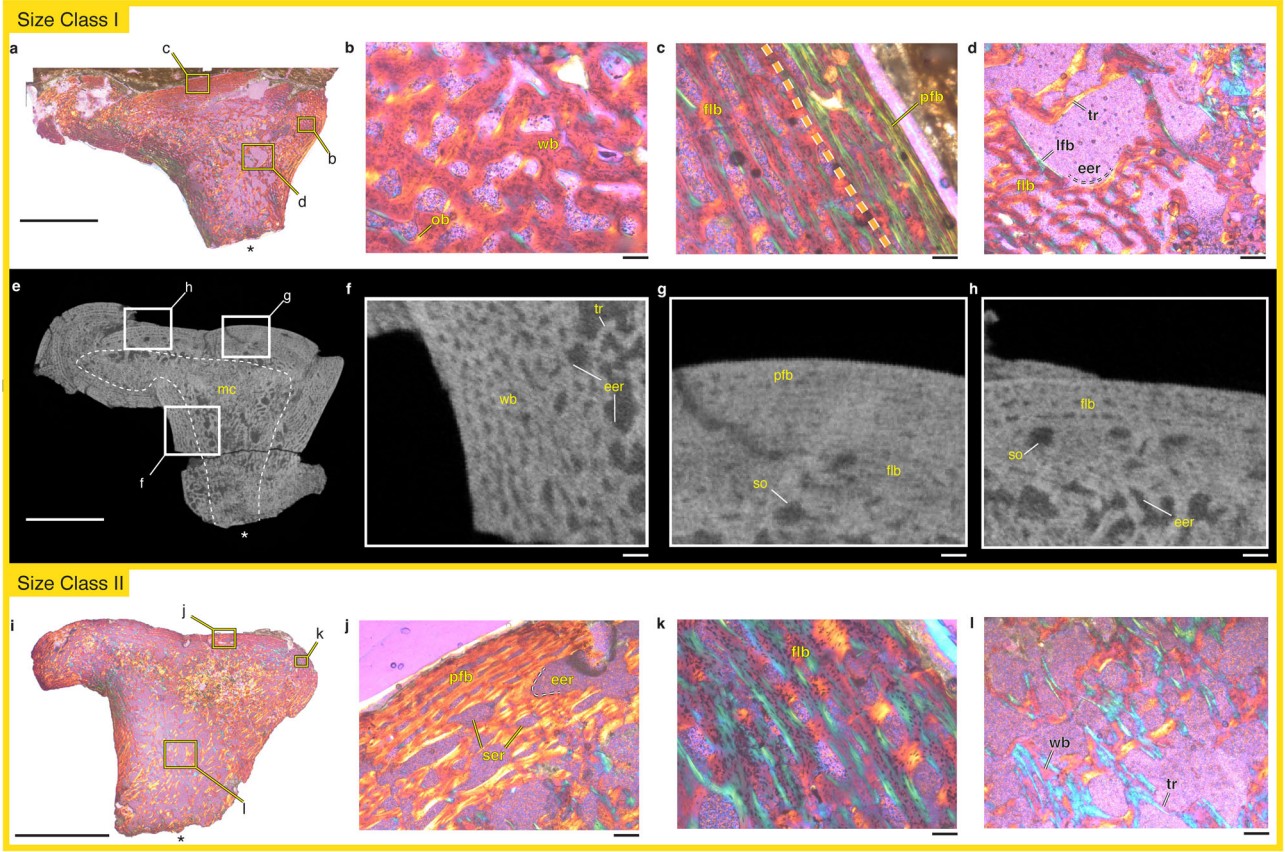

**Fig. 2 The histology of *Whatcheeria* femora from size classes I and II.** Under polarized light and a lambda filter, FMNH PR 5022 represents size class I (**a–d**) and contains a comparatively thick cortex (**a**) composed of fibrolamellar bone (**b–d**) and some parallel-fibered bone at the periosteal surface (**c**). Expansion of the medullary cavity results in endosteal remodeling of the fibrolamellar cortex (**d**). μCT scans of an additional size class I specimen, FMNH PR 1735, display similar tissue distribution (**e**) and composition (**f–h**). FMNH PR 5021 represents size class II (**i–l**). Although there are endosteally located sections of fibrolamellar bone (**k, l**), the narrow cortex of this specimen is mostly composed of parallel-fibered bone that is actively being remodeled both from the endosteal surface and via secondary remodeling (**j**). In **a, e, i**, the adductor crest (ventral) is oriented towards the bottom and marked with an asterisk. Scale bars: a, e, i = 5 mm; **d, f, g, h, k** = 250 μm; b, c, j = 100 μm. Abbreviations: eer endosteal erosion, flb fibrolamellar bone, ob osteonal bone, pfb parallel-fibered bone, ser secondary remodeling erosion, so secondary osteon, tr trabecular bone, wfb woven-fibered bone.

(e.g. FMNH PR 5022) is characterized by a thick cortex composed primarily of rapidly deposited fibrolamellar bone (Figs. 2a–d, 5a). However, the periosteal surface of the cortex contains more slowly deposited parallel-fibered bone, and there is increasingly active endosteal remodeling. The slowdown of periosteal deposition and active endosteal remodeling cause an abrupt thinning of the cortex by size class II (e.g. FMNH PR 5021), with an almost complete reworking of the fibrolamellar tissue (Figs. 2i–l, 5b), and development of trabecular struts. Periosteal bone deposition continues to slow alongside active endosteal remodeling, ultimately resulting in a very thin cortex formed of lamellar bone in size classes III and IV (e.g. FMNH PR 1962, FMNH PR 5023; Figs. 3, 4, 5c, d), with only small remnants of parallel-fibered bone remaining and an extensive trabecular network. Based these data, we correlate the size classes to the following ontogenetic stages: size class I = late-stage juvenile; size class II = sub-adult; size class III = adult; size class IV = skeletally mature adult.

The critical changes to bone tissue organization throughout the sub-adult to adult phases of *Whatcheeria* femoral ontogeny result from a complete reworking of the primary juvenile tissue. Therefore, study of *Whatcheeria* adult stage histology alone would not include the primary fibrolamellar bone that was rapidly deposited at the earliest stages of ossification and persisted until reaching nearly three-fourths of adult size. Further, in the

late-stage juvenile specimen (FMNH PR 5022)—which represents the smallest known size class for *Whatcheeria*—parallel-fibered bone is found at the periosteal surface. This signifies a reduction in growth rate associated with late juvenile/sub-adulthood and thus, this specimen by no means represents the earliest stages of ossification for *Whatcheeria*. Our finding calls into question why younger juveniles are not recovered from an otherwise abundantly fossiliferous locality. It may be that a short juvenile phase simply reduces the probability of being preserved in the fossil record. Alternatively, there could be a body size bias against younger/smaller individuals at the Hiemstra Quarry because of taphonomic sorting[57]. Finally, another possibility is that young juveniles occupied distinct habitats or nurseries[24,58] and therefore small individuals were not captured alongside the sub-adult and adult population. In any scenario, the absence of young juveniles is curious and important to note in the record of *Whatcheeria* and other early tetrapod ontogenetic series[37].

**Life history and ecology**. The elevated juvenile growth rates associated with reticular fibrolamellar bone may have served as an adaptation to the particular ecology and life history of *Whatcheeria*: individuals could have reached maximum body size and sexual maturity more quickly[5], allowing them to occupy a large-bodied, predatory niche in their paleoecosystem[44,59]. Within its

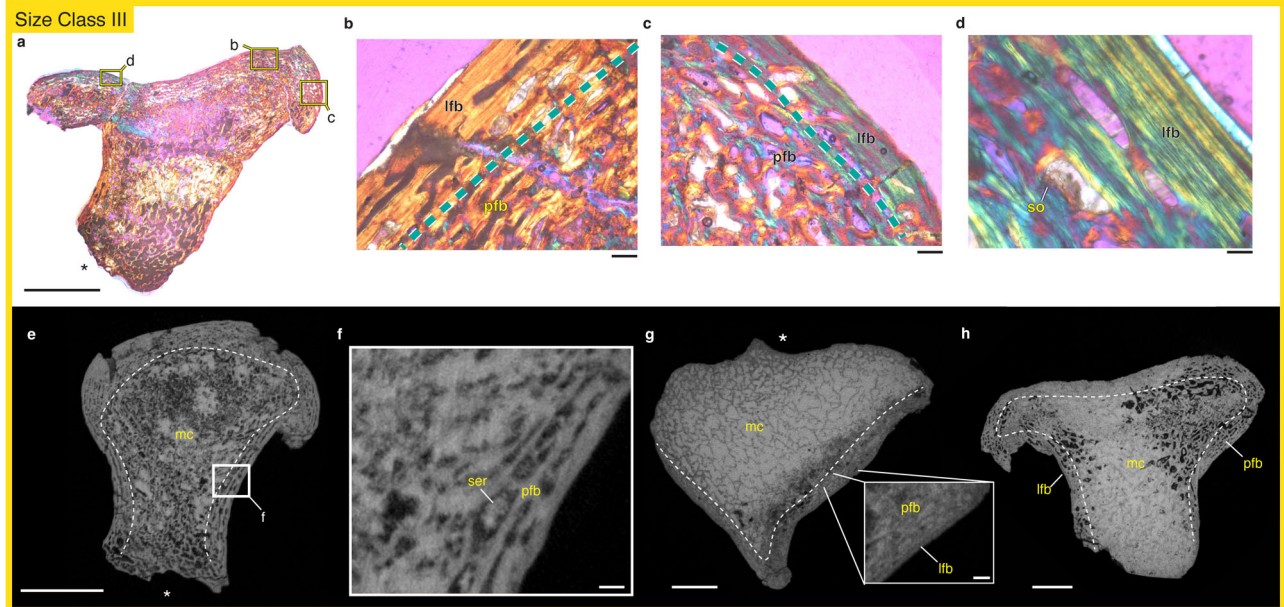

**Fig. 3 The histology of *Whatcheeria* femora from size class III.** Thin sections of FMNH PR 1962 visualized under polarized light and a lambda filter (**a**–**d**) reveal a cortex composed of both lamellar and parallel-fibered bone (**b**–**d**). There is evidence of ample secondary remodeling in this specimen (**d**). μCT scans of a slightly smaller femur, FMNH PR 1952, contain evidence of a largely parallel-fibered cortex and little lamellar bone (**e**, **f**) suggesting this specimen was at the earliest stages of growth in size class III. Additional specimens within size class III include FMNH PR 1992 (**g**) and FMNH PR 1760 (**h**), both of which have narrow cortices composed primarily of what appears to be parallel-fibered bone with lamellar bone along the periosteal surface. In **a**, **e**, **g**, **h** the adductor crest (ventral) is oriented towards the bottom and marked with an asterisk. Abbreviations: lfb lamellar-fibered bone, mc medullary cavity, pfb parallel-fibered bone, ser secondary remodeling erosion, so secondary osteon. Scale bars: **a**, **e**, **g**, **h** = 5 mm; **b**, **c**, **f**, inset **g** = 250 μm; **d** = 100 μm.

lowland terrestrial lake system, *Whatcheeria* is recovered as the largest-bodied tetrapod (1.5–2 m) with cranial adaptations indicating that it consumed other large-bodied vertebrates via forceful biting[45]. Further, geologic data suggest that populations of *Whatcheeria* lived during times of global climate change[60] and local increases in wet-dry seasonality[59–61]. These unpredictable global and/or local environmental conditions may have further promoted a selective regime favoring a rapid juvenile phase, reducing time to reproductive age[19,21,62].

The pattern of growth described here for *Whatcheeria* strongly juxtaposes that recently described for a slightly younger Carboniferous stem tetrapod, *Greererpeton*[37], whose elongated and gracile gross anatomy suggest a particularly aquatic, perhaps benthic lifestyle[63] (Fig. 5). An ontogenetic dataset showed that *Greererpeton* growth was characterized by a moderately paced rate of bone deposition, cortical thickening through development, and some evidence of growth marks[37] (Fig. 5e–g). Although *Whatcheeria* and *Greererpeton* were not contemporaries, this divergence in growth pattern may reflect the occupation of different ecological niches—with *Whatcheeria* considered to have been an active, large-bodied macropredator[44] that grew rapidly without interruption into sub-adulthood and *Greererpeton* employing a slow-and-steady, punctuated growth pattern that may reflect its more benthic lifestyle and less active mode of predation[37,63,64]. Importantly, these two ontogenetic histological series reveal that Carboniferous tetrapods employed a range of life history strategies that may have been consistent with the particulars of their ecological roles and habitats.

The narrow adult cortices of *Whatcheeria* are comparable to highly aquatic crown amniotes that are dynamic swimmers who employ active means of buoyancy control (e.g. elephant seals, ichthyosaurs, and whales)[64]. Although these similarities support the possibility that *Whatcheeria* was an active aquatic predator and dynamic swimmer, such comparisons must be approached

with a great deal of caution given the likely compounding differences in ecology and phylogeny. Crown group amniotes that display bone histology similar to *Whatcheeria* are secondarily aquatic and pelagic whereas *Whatcheeria* is plesiomorphically aquatic/semi-aquatic and lived in a terrestrial lake setting[59]. The similarities that do exist between *Whatcheeria* and pelagic predators warrants further investigation. However, they also highlight the many difficulties in drawing direct comparisons between stem tetrapods and crown amniote groups.

*Whatcheeria* also deviates from the 'standard' slow growth and extended juvenile phase interpreted for Devonian stem tetrapods and tetrapodomorph fish[24,25,33,49,65,66]. For instance, it is posited that *Acanthostega* maintained an unossified, cartilaginous humerus until late juvenile stage, after which lamellar bone was deposited slowly to form a narrow cortex surrounding an expansive medullary cavity filled with trabeculae[24]. Interestingly, the narrow lamellar cortices and well-developed trabeculae of sub-adult/adult *Whatcheeria* specimens closely resemble the bone tissue microstructure described from *Acanthostega* and other humeral and femoral samples of Devonian taxa[24,25,33,48]. This finding implies that a similar adult bone tissue composition may be achieved via two very different life history strategies. However, as the primary juvenile tissues are almost completely remodeled in *Whatcheeria* by sub-adulthood, it is possible that the noted differences in life history strategies are merely due to missing substantial parts of the bone growth record from Devonian stem tetrapods. Considering the rarity of Devonian fossils, and especially ontogenetic series, distinguishing between these two possible scenarios remains challenging at present.

Likewise, the deposition of fibrolamellar bone during the growth phase of *Whatcheeria* requires a reconsideration of modern analogs for studying the life histories of stem tetrapods. Although the presumed amphibious anatomy and behavior of

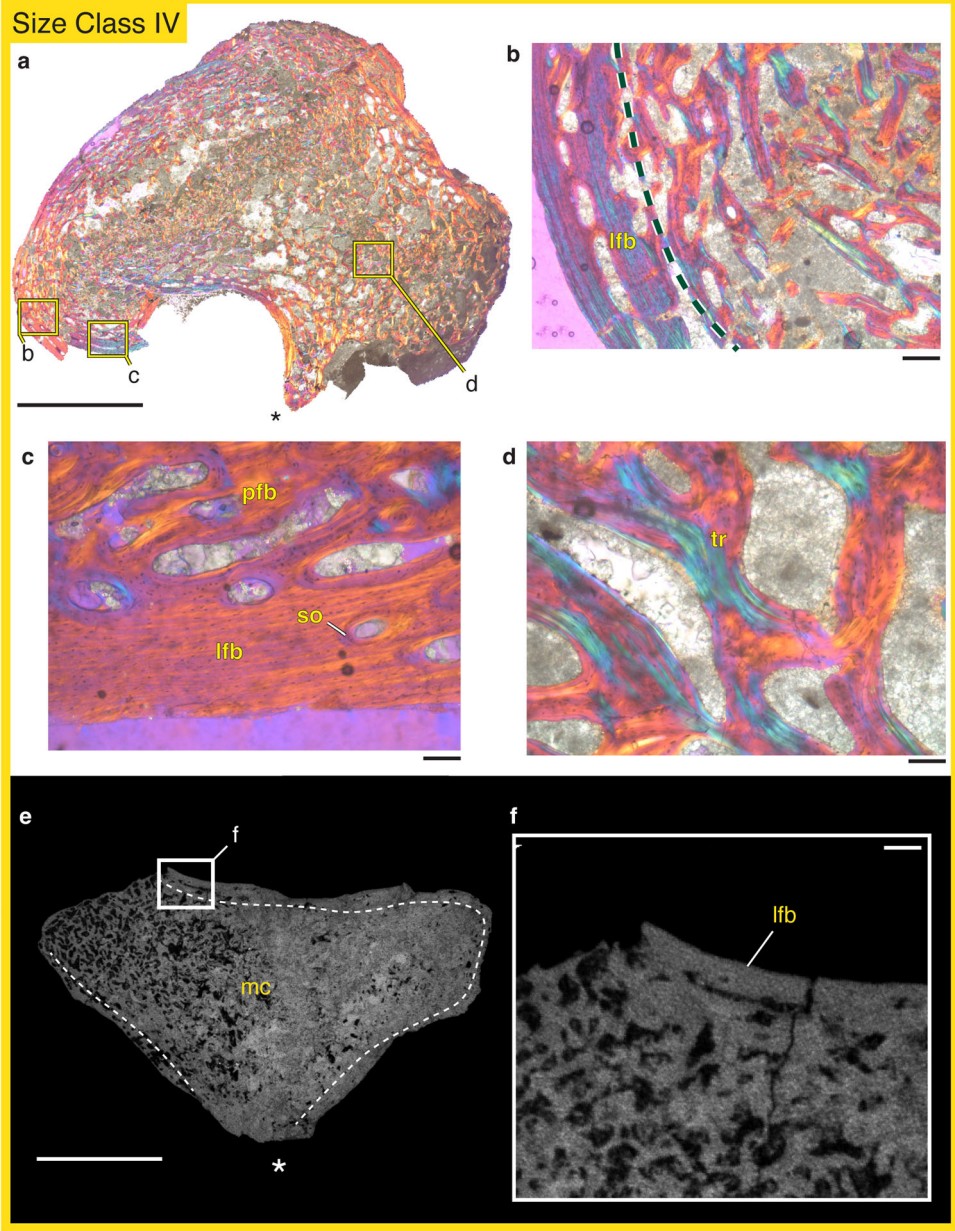

**Fig. 4 The histology of *Whatcheeria* femora from size class IV.** Under polarized light and a lambda filter, FMNH PR 5023 contains an abundant trabecular network in the medullary cavity and a narrow cortex (**a**). The cortex is composed primarily of lamellar bone (**b**, **c**) with some secondary remodeling (**c**). The trabeculae in this specimen are well-developed and composed of lamellar tissue (**d**). These general observations are consistent with an additional specimen, FMNH PR 1958, visualized with μCT imaging (**e**), including a developed trabecular network in the medullary cavity and a narrow lamellar cortex (**f**). In **a** and **e** the adductor crest (ventral) is oriented towards the bottom and marked with an asterisk. Abbreviations: lfb lamellar-fibered bone, mc medullary cavity, pfb parallel-fibered bone, so secondary osteon, tr trabeculae. Scale bars: **a**, **e** = 5 mm; **b**, **f** = 250 μm; **c**, **d** = 100 μm.

stem tetrapods commonly draws natural comparisons with modern and fossil amphibians[25,67,68], factors such as large body size in Devonian-Carboniferous tetrapods[69] and the highly derived life cycles of modern amphibians[70] are rarely recognized as potentially compounding influences on life history comparisons. Whereas some stem tetrapods may have employed a slow-and-steady growth pattern comparable to that typically attributed to modern amphibians, the rapid juvenile growth described here in *Whatcheeria* deviates from this stereotyped pattern. This suggests that amphibians may not serve as a 'broad spectrum' analog to all Devonian and Carboniferous tetrapods and more careful consideration for the diversity of potential life histories is critical in making inferences in the fossil record.

## Conclusions

We document a remarkably early occurrence of fibrolamellar bone in tetrapod evolution, expanding the temporal and phylogenetic range of rapid juvenile growth rates outside crown tetrapods. Elevated juvenile growth could have allowed *Whatcheeria* to reach skeletal and reproductive maturity more rapidly and to attain a larger adult body size, supporting an ecological role as a top predator in its paleoenvironment. The life history strategy of *Whatcheeria* contrasts with fellow Mississippian stem tetrapod *Greererpeton* as is evident in the distinct organization of their bone tissues throughout ontogeny, and this difference likely reflects divergent ecological roles. Although adult *Whatcheeria* bone tissue microstructure appears similar to that described in

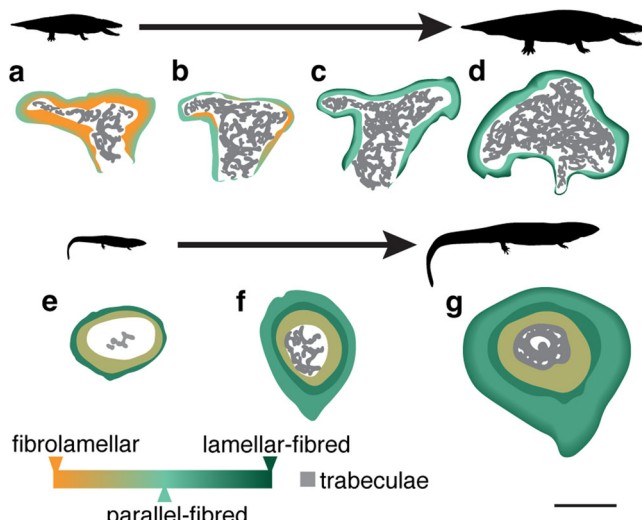

**Fig. 5 Comparative schematic of the organization of bone tissue in two Carboniferous stem tetrapods, *Whatcheeria* and *Greererpeton*, throughout ontogeny.** *Whatcheeria* (**a**–**d**) is a large-bodied predator whose juvenile growth is characterized by fibrolamellar bone (**a**) that is eventually remodeled and completely replaced with parallel-fibered tissue (**b**, **c**) and subsequent slow deposition of lamellar bone (**c**, **d**). By contrast, *Greererpeton*[37] (**e**–**g**) is characterized by moderately paced bone deposition, even early in ontogeny (**e**), with a distinct lamellar band of bone deposited at the late juvenile stage (**e**–**g**). Sub-adult growth in *Greererpeton* is subsequently characterized by slowly deposited tissues (**f**) as well as endosteal deposition that results in a particularly thick adult cortex (**g**). Scale bar = 5 mm.

Devonian taxa, it remains unclear whether this similarity is an artifact of sampling or achieved convergently through distinct modes of growth. Continued investigations into patterns of growth through ontogeny in both Devonian and Carboniferous tetrapods will help shed light on the diversity of life history strategies employed during early tetrapod evolution and the potential physiological underpinnings of the conquest of land.

## Methods

**Geological setting.** All *Whatcheeria* specimens come from the now-defunct Jasper Hiemstra Quarry, Keokuk County, Iowa (Fig. 1b). The Hiemstra Quarry preserves a mixture of marine and terrestrial sedimentary rocks that can be correlated with the St. Genevieve Formation of the Illinois Basin on the basis of sequence stratigraphy, with an age of 331–326 Mya (late Viséan–early Serpukhovian)[59]. Tetrapod fossils at the Hiemstra Quarry are preserved in a unit comprised of interbedded limestone conglomerates and dark laminated shales that fills two depressions or collapse structures in the underlying marine limestone[59,71]. The interpreted depositional environment is a sinkhole that formed in association with the drying up of a lowland terrestrial lake, with fossils accumulating during influxes of debris from the surrounding environment[59,71].

**Systematics.** *Whatcheeria deltae*[72] is a tetrapod that belongs to the family Whatcheeriidae[73] alongside various other taxa (see Otoo et al.[44] for a review). We note there is debate concerning which fossil taxa are considered tetrapods *sensu stricto*, with some using a crown group definition and others using a total group definition that includes all tetrapodomorphs that have limbs with digits as opposed to fins with lepidotrichia[74]. In this paper, we classify all taxa with limbs and digits as tetrapods and those that fall stemward of the crown group as stem tetrapods, including *Whatcheeria*.

**Specimen selection.** We sampled nine femora that were previously collected on private property between 1985 and 1988. These specimens are accessioned at the Field Museum of Natural History (FMNH PR 1735, PR 1760, PR 1952, PR 1958, PR 1962, PR 1992, PR 5021, PR 5022, PR 5023) and have been referred to *Whatcheeria* based on the criteria outlined in Otoo et al.[44], in particular: the lack of internal trochanter, thick adductor blade, and broad fourth trochanter (Supplementary Fig. 1). All of our histologically sectioned specimens experienced taphonomic dorsoventral compression, obscuring the original cross-sectional shape.

However, components of the cortex and medullary cavity remain intact and provide clear data on the histology of both bony regions. Given intraskeletal variation in the record of growth and histological signatures[64,75–77], femora were exclusively sampled in this study to maintain a consistent comparison throughout ontogeny. It is worth noting that our data and interpretations relate exclusively to femoral histology.

**Histological analyses.** All nine specimens were scanned using high-resolution micro-computed tomography (μCT) to create virtual thin sections, as well as to preserve whole-bone anatomy prior to destructive sampling. Specimens were scanned using a Bruker Skyscan 1173 μCT system in the Museum of Comparative Zoology (MCZ) Digital Imaging Facility. All specimens were scanned at 130 kV voltage and 61μA current using a 0.25 mm brass filter and all scan data are archived at FMNH. Destructive histological thin-sectioning was conducted on four specimens that represented the four size classes (I–IV) outlined in Otoo et al.[44]. Standard thin-sectioning protocols[78] were applied to produce transverse thin-sections at the mid-diaphysis of the femora of FMNH PR 5022, PR 5021, PR 5023, and PR 1962. The mid-diaphysis of long bones capture the lengthiest record of growth as they serve as the primary center of ossification[79]. Thus, mid-diaphyseal transverse cross-sections of long bones serve as the standard in paleohistological life history analyses[78]. Total cross-sectional and cortical bone areas were measured in ImageJ[80] to describe the percentage of cortical bone contribution to cross-sectional area. These values are provided in Supplementary Table 1.

**Reporting summary.** Further information on experimental design is available in the Nature Research Reporting Summary linked to this paper.

## Data availability

All histological images have been uploaded as single, high-resolution images to MorphoBank for widespread availability of the data presented here. Access to these images can be found in MorphoBank under Project P4272:Ontogenetic histology of *Whatcheeria deltae*. Images will be published online at time of publication.

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

## Acknowledgements

We thank S. Johnston for additional preparation and preservation of the specimens used in this study. W. Simpson, A. Stroup, and C. Byrd assisted with specimen loans and curation. Finally, we acknowledge J. Bolt whose team conducted the fieldwork that led to the discovery and collection of *Whatcheeria deltae*.

## Author contributions

M.R.W. conducted histological sectioning, data analysis, figure preparation, and wrote the manuscript. B.K.A.O. identified samples, contributed taxonomic expertise, and assisted with manuscript development. K.D.A. supervised specimen allocation and assisted with manuscript development. S.E.P. conceptualized and oversaw the implementation of the project, funded the project, and wrote the manuscript.

## Competing interests

The authors declare no competing interests.
