## [Peer Review File · Communications Biology]

Reviewers' comments:

Reviewer #1 (Remarks to the Author):

This manuscript entitled "Fossil bone histology reveals ancient origins for rapid juvenile growth in tetrapods" examined changes in bone histology along a series of body sizes in the stem tetrapod *Whatcheeria deltae*, and found the bone tissue characteristic to rapid growths in small (i.e., young) individuals, eventually highlighting the diversity of ontogenetic strategies in the early tetrapods for the first time. The main conclusions are convincing and impressive. Therefore, I believe this study is worth publishing from Communications Biology, although there is room for improvement before publication.

Major comments.

(1) In the current version of the manuscript, the information about the positions of cross-sections is not provided in Figures 2–4 and Supplementary Fig. 1. In Methods section, the position is specified only as "the mid-diaphysis." Since such data are crucial for rigorous comparisons of bone histology, in particular cortical thickness, the planes of cross-sections should be clearly described. In addition, the reason(s) to chose the sectioned planes presented in this study should be explained, as well. Some distinct ridges are recognizable in the cross-sections, and I suppose some of them correspond to anatomical structures, such as the fourth trochanter. Comparisons should be conducted, ideally, between homologous position, and labels of anatomical structures on the cross-sections are necessary for showing the positions to be compared.

(2) The method for calculating ratios of cortical bones in cross-sectional areas should be explained. Perhaps, a certain image analysis software was used for the calculations? In general, given that a thickness of cortical bone is unchanged through growth, the ratio of cortical bone in cross-sectional area decreases as the diameter of the long bone increases. The decrease in cortical area ratio from FMNH PR 5022 (57%) to PR 5023 (35%) could happen even under a condition, where the thickness of cortical bone was unchanged (assuming that PR 5023 was >1.8 times larger than PR 5022).

Indeed, I agree with the authors on that the thickness of cortical bone decreases through growth, as such a change is recognizable in Figs. 2–4, but the measured values should be provided, rather than (or along with) the area ratios. Although the authors argue that "bone tissue organization and cortical thickness change with size class" in the manuscript (page 6, line147), the change in cortical thickness is unclear in the current version of the manuscript.

(3) As μ CT data of different size classes were obtained in this study, histological observations of the epiphyses may be feasible, and I am wondering if the maturities of individuals can be evaluated to a certain degree based on histological features of the epiphyses.

(4) According to Otto et al. (2021: ref. 42), individuals of the size class I were "probably smaller than a meter in length." It is likely that the body size at hatching was much smaller than that of the class I, and data about early stages of growth are currently lacking. Therefore, the discussion about a selective advantage of the life history of *Whatcheeria* (page 3, line 61; pages 7–8, lines 178–186) sounds only speculative.

Minor comments.

(5) Figure 1.

The silhouettes appear to be provided from image banks (e.g., PhyloPic). If so, please re-check the credits of the images. In addition, it is better to change the orientation of the silhouette of the amphibian in accordance with the other silhouettes.

(6) Data of μ CT were currently not deposited in MorphoBank, thus the explanation that "All histological images have been uploaded" (page, 11, Method section) is not necessarily adequate. Ideally, the μ CT data should be deposited in a public repository (e.g., MorphoSource), but if impossible, an explanation about the data availability of μ CT data should be provided in Data Availability section.

Reviewer #2 (Remarks to the Author):

Dear Editor, Dear Authors,

Please find enclosed my review for the manuscript " Fossil bone histology reveals ancient origins for rapid juvenile growth in tetrapods" (COMMSBIO-22-2182) submitted to Communications Biology, along with an annotated PDF with additional comments on specific sections of the paper.

This paper presents the first description of femoral bone histology in an ontogenetic series of *Whatcheeria deltae*, an early-diverging tetrapodomorph from the Early Carboniferous of Iowa. The authors have sampled nine specimens encompassing all four ontogenetic stages described for this species in previous literature, and provide a comprehensive description of its growth pattern. The description is sound, well-illustrated, and showcases the unique bone microstructure of this species. The main result of the paper is the presence, in early ontogenetic stages, of a layer of fibrolamellar bone (FLB), a bone tissue known to be associated with high bone growth rates, hitherto undocumented in early tetrapodomorphs and extant lissamphibians. The authors discuss the implication of this finding in the context of ancestral bone growth and metabolism at the base of tetrapods, and show that extant lissamphibians may not be as good a proxy for the ancestral condition of these traits as previously thought.

I agree with most of the authors' hypotheses and conclusions. This is indeed an unexpected result that has very important implications for the evolution of fast bone growth rates in early tetrapods, and yet another example of such a bone growth pattern having been acquired at a much more inclusive level of the vertebrate tree than previously expected. The methods are adequately described and the results well presented. I do, however, disagree with the way the authors use these histological results to hypothesize high metabolic rate in this species, and potentially in other tetrapodomorphs as well. The correlation between high bone growth rate and endothermy is notoriously hard to establish for any given vertebrate clade (see annotated PDF for more details and references), and since this study does not present data to confirm or infirm such a pattern on a sample of tetrapodomorphs, the manuscript as it is jumps to conclusions on heat physiology without direct evidence. The authors should carefully revise these aspects of the manuscript and be much more cautious about this interpretation of their results.

Another issue I have with the discussion is the lack of comparison with several relevant clades of aquatic tetrapods. The authors have rightfully mentioned another previously described Carboniferous tetrapodomorph with a drastically different bone histology for comparison, but the histology of adult *Whatcheeria* is actually very similar to that of large open-water predatory amniotes, such as whales or ichthyosaurs (see annotated PDF for references). Similarly, the presence of FLB in the bones of juvenile specimens has been described in several species of extant crocodylians (see annotated PDF for references) – another clade of ectothermic aquatic predators. Including comparisons of *Whatcheeria* with these taxa and discussing them in the context of the evolution of aquatic lifestyles among stem and crown tetrapods would be highly relevant to this paper and make a better case for the importance of its results.

Finally, it is important to note that vertebrate bone microstructure usually shows very high intraspecimen variability, and inference of growth pattern for any given species cannot be readily obtained from the ontogenetic series of a single bone (e.g. Cullen et al., 2021; Griffin et al., 2021). Since their sample only includes femora, the authors should mention in their discussion that the high femoral growth rate they recovered needs to be confirmed by sampling other bones in the skeleton before reaching any conclusions on the global growth pattern of *Whatcheeria*.

A number of other minor remarks are listed in the annotated PDF, including adjustments to phylogenetic terminology and clarifications on the definitions of some terms.

Overall, I recommend major revisions before publication – once these points are clarified, this manuscript will be a significant addition to paleohistological literature, and I look forward to seeing it published.

References

- Cullen, T.M., Brown, C.M., Chiba, K., Brink, K.S., Makovicky, P.J., Evans, D.C., 2021. Growth variability, dimensional scaling, and the interpretation of osteohistological growth data. *Biology Letters* 17, 20210383.
- Griffin, C.T., Stocker, M.R., Colleary, C., Stefanic, C.M., Lessner, E.J., Riegler, M., Formoso, K., Koeller, K., Nesbitt, S.J., 2021. Assessing ontogenetic maturity in extinct saurian reptiles. *Biological Reviews* 96, 470–525.

We thank both reviewers for the helpful and thoughtful comments on our manuscript. We agree with their suggestions and have accepted the vast majority of the comments with only a few misunderstandings that we have looked to clarify. Below, we detail our point-by-point response (in red) to the comments from both reviewers.

Reviewer 1

Major comments.

(1) In the current version of the manuscript, the information about the positions of cross-sections is not provided in Figures 2–4 and Supplementary Fig. 1. In Methods section, the position is specified only as "the mid-diaphysis." Since such data are crucial for rigorous comparisons of bone histology, in particular cortical thickness, the planes of cross-sections should be clearly described.

In addition, the reason(s) to chose the sectioned planes presented in this study should be explained, as well. Some distinct ridges are recognizable in the cross-sections, and I suppose some of them correspond to anatomical structures, such as the fourth trochanter. Comparisons should be conducted, ideally, between homologous position, and labels of anatomical structures on the cross-sections are necessary for showing the positions to be compared.

We have added additional language and citations in the Methods to clarify why we used the standard histological protocol of sectioning a transverse plane of the mid-diaphysis when examining the growth record, and the rationale for this choice. To help identify comparable structures, we have added text to the figure legend pointing out the homologous adductor crest present on the ventral aspect of *Whatcheeria femora*.

(2) The method for calculating ratios of cortical bones in cross-sectional areas should be explained. Perhaps, a certain image analysis software was used for the calculations? In general, given that a thickness of cortical bone is unchanged through growth, the ratio of cortical bone in cross-sectional area decreases as the diameter of the long bone increases. The decrease in cortical area ratio from FMNH PR 5022 (57%) to PR 5023 (35%) could happen even under a condition, where the thickness of cortical bone was unchanged (assuming that PR 5023 was >1.8 times larger than PR 5022).

Indeed, I agree with the authors on that the thickness of cortical bone decreases through growth, as such a change is recognizable in Figs. 2–4, but the measured values should be provided, rather than (or along with) the area ratios. Although the authors argue that "bone tissue organization and cortical thickness change with size class" in the manuscript (page 6, line147), the change in cortical thickness is unclear in the current version of the manuscript.

We agree that providing more details on these calculations is needed. We have provided additional text in the Methods section and a supplementary table with the raw values for total cross-sectional area and cortical bone cross-sectional area. From these two values we determine the percent of total area that was composed of cortical bone.

(3) As μ CT data of different size classes were obtained in this study, histological observations of the epiphyses may be feasible, and I am wondering if the maturities of individuals can be evaluated to a certain degree based on histological features of the epiphyses.

Although the epiphyses of amniotes can provide insight into relative maturity, this is unfortunately not true in early tetrapods. The epiphyses remain histologically consistent through development. See Whitney and Pierce (2021) for an example of the stereotyped patterns of

early tetrapod epiphyses.

(4) According to Otto et al. (2021: ref. 42), individuals of the size class I were "probably smaller than a meter in length." It is likely that the body size at hatching was much smaller than that of the class I, and data about early stages of growth are currently lacking. Therefore, the discussion about a selective advantage of the life history of *Whatcheeria* (page 3, line 61; pages 7–8, lines 178–186) sounds only speculative.

We acknowledge that we don't have data on the earliest stages of ontogeny and that we are presenting an interpretation of the selective advantage rapid early growth may have provided in the life history of *Whatcheeria*. We have made minor changes to the text noted in this comment to reiterate that this is an interpretation and not a conclusion. We have developed this interpretation because fibrolamellar bone is the most commonly deposited bone type in vertebrates that employ this life history strategy (page 6, lines 130-131). Although we do not have a record of the earliest stages of ossification, this bone type preserves details of the juvenile growth record that can convey the overall somatic growth pattern during these early stages. The presence of fibrolamellar bone as the youngest tissue indicates that although the absolute earliest stages of growth are missing, growth rates were at least on par with the rapid rates associated with the tissue early in ontogeny. Given that it is the largest predator recovered from its paleoenvironment (page 8, line 181), it is parsimonious to interpret the rapid growth we observed early in *Whatcheeria* ontogeny provided a similar selective advantage to that observed in modern large-bodied predators who grow rapidly to reach adulthood and sexual maturity.

Minor comments.

(5) Figure 1.

The silhouettes appear to be provided from image banks (e.g., PhyloPic). If so, please re-check the credits of the images. In addition, it is better to change the orientation of the silhouette of the amphibian in accordance with the other silhouettes.

Thank you for these comments and we have edited the figure caption to acknowledge PhyloPic and the orientation of the silhouette of the amphibian has been changed. The silhouette of *Ichthyostega* was drawn by us and this is now noted in the figure caption too.

(6) Data of μ CT were currently not deposited in MorphoBank, thus the explanation that "All histological images have been uploaded" (page, 11, Method section) is not necessarily adequate. Ideally, the μ CT data should be deposited in a public repository (e.g., MorphoSource), but if impossible, an explanation about the data availability of μ CT data should be provided in Data Availability section.

Our apologies! We accidentally neglected to release the images. The histological image data should now be available on MorphoBank for the reviewers using the same username and password detailed in the Data Availability Statement. The μ CT data will be archived at the Field Museum as per museum policy, and we have updated the statement. We are also currently working with the Field Museum to see if it might be possible to add our data to MorphoSource.

Reviewer 2

I agree with most of the authors' hypotheses and conclusions. This is indeed an unexpected result that has very important implications for the evolution of fast bone growth rates in early tetrapods, and yet another example of such a bone growth pattern having been acquired at a much more inclusive level of the vertebrate tree than previously expected. The methods are adequately described and the results well presented. I do, however, disagree with the way the authors use these histological results to hypothesize high metabolic rate in this species, and potentially in other tetrapodomorphs as well. The correlation between high bone growth rate and endothermy is notoriously hard to establish for any given vertebrate clade (see annotated PDF for more details and references), and since this study does not present data to confirm or infirm such a pattern on a sample of tetrapodomorphs, the manuscript as it is jumps to conclusions on heat physiology without direct evidence. The authors should carefully revise these aspects of the manuscript and be much more cautious about this interpretation of their results.

In this manuscript, we very much did not want to conclude that *Whatcheeria* was endothermic for the reasons outlined above. Rapid growth and active metabolic rates (& fibrolamellar bone) do not necessarily have to correspond to endothermy. We do acknowledge that some of the language may have particular connotations (e.g. metabolism) and as such have looked to alter the writing such that it reflects our conclusion about growth rates specifically. We hope that making this distinction helps to clarify the limits of our interpretations.

Another issue I have with the discussion is the lack of comparison with several relevant clades of aquatic tetrapods. The authors have rightfully mentioned another previously described Carboniferous tetrapodomorph with a drastically different bone histology for comparison, but the histology of adult *Whatcheeria* is actually very similar to that of large open-water predatory amniotes, such as whales or ichthyosaurs (see annotated PDF for references). Similarly, the presence of FLB in the bones of juvenile specimens has been described in several species of extant crocodylians (see annotated PDF for references) – another clade of ectothermic aquatic predators. Including comparisons of *Whatcheeria* with these taxa and discussing them in the context of the evolution of aquatic lifestyles among stem and crown tetrapods would be highly relevant to this paper and make a better case for the importance of its results.

We think that it is important to note that comparisons between *Whatcheeria* and large open-water predators are prone to a lot of compounding factors that make such comparisons challenging—even if the comparison of bone tissues is similar. *Whatcheeria* was not an open-water predator but rather a lowland terrestrial lake predator. Further, these animals are highly derived compared to *Whatcheeria* as they are secondarily aquatic. That said, we have included text and citations (lines 206-215 of marked version) that include these comparisons. In reference to including a discussion of the evolution of aquatic lifestyles, *Whatcheeria* is a plesiomorphically aquatic animal and thus, does not directly provide evidence on the evolution of aquatic lifestyles in crown tetrapods.

Finally, it is important to note that vertebrate bone microstructure usually shows very high intraspecimen variability, and inference of growth pattern for any given species cannot be readily obtained from the ontogenetic series of a single bone (e.g. Cullen et al., 2021; Griffin et al., 2021). Since their sample only includes femora, the authors should mention in their discussion that the high femoral growth rate they recovered needs to be confirmed by sampling other bones in the skeleton before reaching any conclusions on the global growth pattern of *Whatcheeria*.

This is a very good point! We have added a discussion of this point to the Methods in our section on specimen selection and clarify that these data and subsequent conclusions are based entirely on femoral histology (lines 263-266 of marked version).

Additional responses to the comments made in the annotated PDF. Line numbers refer to original submission.

Line 11: If I'm not mistaken, *Whatcheeria* is not a tetrapod – it is a close outgroup. Please be consistent with phylogenetic nomenclature when referring to clades in this context.

This is a point of debate. Some researchers use a total group (apomorphy-based) definition of tetrapods as those animals having limbs with digits (limbed tetrapodomorphs), while others use a crown group definition. We prefer to use a total group definition. This means that *Whatcheeria*, *Pederpes* and all limbed tetrapods that fall outside the crown are considered stem tetrapods. Since this is a point of contention, we have added justification for our nomenclature usage in the Methods.

Line 30: I assume you refer to the random, isotropic arrangement of collagen fibers in woven bone (e.g. Ricqlès et al., 1991), but here "disorganized" is not very clear. Maybe use another term, e.g. "randomly organized"?

Ricqlès, A. de, Meunier, F.J., Castanet, J., Francillon-Vieillot, H., 1991. Comparative microstructure of bone, in: Hall, B.K. (Ed.), Bone, Volume 3: Bone Matrix and Bone Specific Products. CRC Press, Boca Raton, pp. 1–78.

In this context, disorganized refers to both the randomly organized matrix fibers as well as the haphazard vascular organization. We have modified the text for clarity.

Line 30: This is a bit misleading. Yes, woven bone – and by extension fibrolamellar bone – is indicative of high growth rates, as shown in all major vertebrate groups, but not necessarily of high metabolic rates. In vertebrates, bone growth rates are not necessarily correlated with metabolic rates, since the strength and slope of this correlation can vary a lot depending on the vertebrate clade, and there is considerable overlap between bone growth rates of ectotherms and endotherms (Myhrvold, 2015, 2016). Please clarify.

References

- Myhrvold, N.P., 2016. Dinosaur Metabolism and the Allometry of Maximum Growth Rate. PLOS ONE 11, e0163205. <https://doi.org/10.1371/journal.pone.0163205>
- Myhrvold, N.P., 2015. Comment on "Evidence for mesothermy in dinosaurs." Science 348, 982. <https://doi.org/10.1126/science.1260410>

We want to be very clear about this: we are not making the jump from high metabolic rates to endothermy. There is a correlation between rapid somatic growth rates and the deposition of fibrolamellar bone but that does not necessarily reflect endothermy. It reflects a period of time in the life history of this animal where it experienced elevated growth rates. We have modified the text for clarity.

Line 31: Same as previous comment – fibrolamellar bone (FLB) can also be found in ectothermic taxa in their early ontogeny (e.g. crocodylians – Tumarkin-Deratzian, 2007;

Woodward et al., 2014), while some endothermic mammals and birds with low metabolic rate due to specific life history (e.g. nocturnality, fossoriality, insectivory) can have little to no FLB (e.g. kiwi – Bourdon et al., 2009; Heck and Woodward, 2021; fossorial or nocturnal mammals – e.g. Castanet et al., 2004; Montoya-Sanhueza and Chinsamy, 2017; Legendre and Botha-Brink, 2018). This suggests that the presence/absence of FLB reflects differences in body size, habitat, diet, etc. at least as much, if not more, than differences in bone growth rate. I agree that FLB is likely to be associated with relatively high metabolism, but the correlation is not as tight as suggested here, and it certainly does not correspond to a test of correlation in the statistical sense, as this sentence may suggest. Please correct/clarify.

References

- Bourdon, E., Castanet, J., Ricqlès, A. de, Scofield, P., Tennyson, A., Lamrous, H., Cubo, J., 2009. Bone growth marks reveal protracted growth in New Zealand kiwi (Aves, Apterygidae). *Biology Letters* 5, 639–642. <https://doi.org/10.1098/rsbl.2009.0310>
- Castanet, J., Croci, S., Aujard, F., Perret, M., Cubo, J., de Margerie, E., 2004. Lines of arrested growth in bone and age estimation in a small primate: *Microcebus murinus*. *Journal of Zoology* 263, 31–39. <https://doi.org/10.1017/S0952836904004844>
- Heck, C.T., Woodward, H.N., 2021. Intraskkeletal bone growth patterns in the North Island Brown Kiwi (*Apteryx mantelli*): Growth mark discrepancy and implications for extinct taxa. *Journal of Anatomy* 239, 1075–1095. <https://doi.org/10.1111/joa.13503>
- Legendre, L.J., Botha-Brink, J., 2018. Digging the compromise: investigating the link between limb bone histology and fossoriality in the aardvark (*Orycteropus afer*). *PeerJ* 6, e5216. <https://doi.org/10.7717/peerj.5216>
- Montoya-Sanhueza, G., Chinsamy, A., 2017. Long bone histology of the subterranean rodent *Bathyergus suillus* (Bathyergidae): ontogenetic pattern of cortical bone thickening. *Journal of Anatomy* 230, 203–233. <https://doi.org/10.1111/joa.12547>
- Tumarkin-Deratzian, A.R., 2007. Fibrolamellar bone in wild adult Alligator mississippiensis. *Journal of Herpetology* 41, 341–345.
- Woodward, H.N., Horner, J.R., Farlow, J.O., 2014. Quantification of intraskkeletal histovariability in Alligator mississippiensis and implications for vertebrate osteohistology. *PeerJ* 2, e422. <https://doi.org/10.7717/peerj.422>

Again, we are not making the jump from a period of rapid growth rates in the early ontogeny of *Whatcheeria* to endothermy. We lack sufficient data to conclude endothermy and therefore do not make any conclusions about endothermy. We also agree that FLB can be found during early ontogeny in ectothermic amniotes such as crocodiles. Such bone has traditionally been thought of as an amniote feature, that is why finding it in a stem tetrapod was so surprising!

Line 32: Over the last decade, the definition of FLB has changed considerably, with many previous descriptions of it found to be either ambiguous or incorrect (Stein and Prondvai, 2014). FLB is now considered a special case of woven-parallel complex (Prondvai et al., 2014). While this is a complex subject that the present study is not expected to extensively delve into, this debate deserves at least a mention here (or later in the paper).

References

- Prondvai, E., Stein, K.H.W., de Ricqlès, A., Cubo, J., 2014. Development-based revision of bone tissue classification: the importance of semantics for science. *Biological Journal of the Linnean Society* 112, 799–816. <https://doi.org/10.1111/bij.12323>
- Stein, K., Prondvai, E., 2014. Rethinking the nature of fibrolamellar bone: an integrative biological revision of sauropod plexiform bone formation. *Biological Reviews* 89, 24–47. <https://doi.org/10.1111/brv.12041>

Thank you for this suggestion. We have added additional text that refers directly to this further refined definition of FLB.

Line 36: Do you mean evidence presented in this study, or previously published research? You have not cited references on the bone histology of early amniotes so far, so I am not sure what this current evidence is. Please clarify.

We agree this is confusing and have edited the text to hopefully clarify.

Line 51: Same as for *Whatcheeria* – *Pederpes* is a non-tetrapod tetrapodomorph. Please correct.

Please see our comment above about the use of a total group definition.

Line 131: This is vague – what do you consider to be a large body size in this context? FLB is commonly found in the bone cortices of many small mammals (e.g. rodents) and birds (e.g. songbirds) – see chapters 27 and 29 in Buffrénil et al. (2021). Since it was already present in nonmammalian synapsids and nonavian dinosaurs, it likely reflects phylogenetic signal more than it does body size.

Buffrénil, V. de, Zylberberg, L., Ricqlès, A.J. de, Padian, K. (Eds.), 2021. Vertebrate Skeletal Histology and Paleohistology. CRC Press, Boca Raton. <https://doi.org/10.1201/9781351189590>

We agree that this sounds more vague than we intended. We have removed the comment on body size here.

Line 141: Again, this is a very strong implication that the data shown here does not support directly, and that has been extensively debated in the literature. Identifying FLB in the early growth stages of *Whatcheeria* is evidence for high bone growth rates in early ontogeny, not for high metabolic rate. This does not prevent you from discussing such hypotheses here, but you have to be much more cautious about such an interpretation and mention relevant literature regarding the link between bone microstructure and physiology across vertebrates.

We agree that physiology does suggest untested conclusions (e.g. endothermy). As such, we have changed the language throughout the manuscript to reflect growth rather than physiology.

Line 171: Not sure what you mean by this – I get that juveniles likely grew and reached the subadult stage relatively fast, but there will always be a moment right after hatching where there will be a high proportion of juveniles among the population, regardless of how fast they grow later on. This could make the discovery of juvenile specimens less likely overall in the present time, but that does not mean that the population would present fewer juveniles at 'any given moment'.

We agree that the language on this is confusing and have altered the text to make our point clearer.

Line 172: Maybe include a citation for this.

Good suggestion and we have added a citation (Behrensmeyer 1978).

Line 192: "Highly aquatic" is confusing here – are you suggesting that the histological pattern documented here for *Whatcheeria* is not typical of aquatic animals? An aspect that you do not discuss here and that would be fascinating to explore is that the pattern you describe in adult *Whatcheeria* specimens – a very thin cortex of lamellar bone that surrounds very dense trabecular bone that represents most of the cross-section – is typical of large, active aquatic predators, e.g. cetaceans and ichthyosaurs (Houssaye et al., 2015, 2016). It seems to me that the difference in pattern between *Whatcheeria* and *Greererpeton* could be explained by a difference in predation strategy – active open-water hunting for the former, sit-and-wait ambush for the latter. Discussing the implications of such differences would make a stronger case for the importance of your results in the context of amniote bone histology and evolution.

References

- Houssaye, A., Sander, P.M., Klein, N., 2016. Adaptive Patterns in Aquatic Amniote Bone Microanatomy—More Complex than Previously Thought. *Integrative and Comparative Biology* 56, 1349–1369. <https://doi.org/10.1093/icb/icw120>
- Houssaye, A., Tafforeau, P., de Muizon, C., Gingerich, P.D., 2015. Transition of Eocene Whales from Land to Sea: Evidence from Bone Microstructure. *PLOS ONE* 10, e0118409. <https://doi.org/10.1371/journal.pone.0118409>

Highly aquatic here refers to the fact that the anatomy of *Greererpeton* reflects an especially aquatic lifestyle (elongated trunk and tail; gracile and diminutive limbs) – not that this animal was an open water pelagic predator similar to whales or ichthyosaurs. We agree that our phrasing is confusing and have edited text to clarify. We also agree that the discussion topic of active predator vs. sit and wait predation is worth including and have added text here.

Line 198: Why not just say "Devonian tetrapodomorphs"? "Fish" means nothing in this context.

See comment from abstract.

References: I noticed a few typos in the references – easy to fix, but please check.

Thank you for bringing this to our attention, we have edited these typos.

REVIEWERS' COMMENTS:

Reviewer #1 (Remarks to the Author):

The revised version of the manuscript entitled "Fossil bone histology reveals ancient origins for rapid juvenile growth in tetrapods" has been improved from the previous version, and I am satisfied with the authors' responses. However, there is one point to be clarified, concerning the data newly presented in the revised version, before publication.

Major comment.

(1) According to the measured values of cross-sectional areas newly provided in Supplementary Table 1, the femur of the Size Class I individual FMNH PR 5022 is stouter (20% larger in cross-sectional area) than that of the Class II individual FMNH PR 5021. On the other hand, in the original description of the size classes by Otoo et al. (2021), "the femur [of the class II] is similar to that of Class I but correspondingly larger in size (p. 731)", being seemingly incongruent with the relationship between FMNH PR 5022 and 5021. Some explanation is required regarding this problem in Results part (lines 115–118) and/or Discussion part (lines 154–157).

Minor comments.

(2) Supplementary Fig. 1.

I recommend the authors to indicate the planes of cross sections as lines on the photos of the entire specimens. It will help readers to understand the positions of the cross sections.

(3) Supplementary Table 1.

The caption should be placed above the table.

Reviewer #2 (Remarks to the Author):

Dear Editor, Dear Authors,

Please find enclosed my review for the manuscript "Fossil bone histology reveals ancient origins for rapid juvenile growth in tetrapods" (COMMSBIO-22-2182A) submitted to Communications Biology.

This is the second round of reviews for this manuscript. The authors have done a great job incorporating the suggestions of both reviewers in this new version – uncertainties regarding ontogeny, intraspecific variability, and potential similarities in growth patterns with other tetrapods are now discussed with corresponding references, in a way that allows the reader to look for more information if necessary. I am also really happy with the added systematics subsection of the Methods, which clarifies terminological ambiguities regarding the affinities of *Whatcheeria* among tetrapods and close outgroups. Finally, the authors have carefully edited aspects of their work related to metabolic inference, which makes the discussion on growth patterns in the context of metabolic heat production much more cautious. Overall, I see no further issues with this version – this is an excellent paper that deserves to be accepted as such in Communications Biology. Congratulations to the authors, and thanks for taking the time to answer our comments and queries.

We thank both reviewers for their second look at our manuscript. We find their support of our manuscript exciting and thank both for their time and effort in bringing this work to its final revisional stage.

There are only a few comments from the reviewers and we have addressed them here in a point-by-point response.

Reviewer 1:

“The revised version of the manuscript entitled "Fossil bone histology reveals ancient origins for rapid juvenile growth in tetrapods" has been improved from the previous version, and I am satisfied with the authors' responses. However, there is one point to be clarified, concerning the data newly presented in the revised version, before publication.”

“Major comment.

(1) According to the measured values of cross-sectional areas newly provided in Supplementary Table 1, the femur of the Size Class I individual FMNH PR 5022 is stouter (20% larger in cross-sectional area) than that of the Class II individual FMNH PR 5021. On the other hand, in the original description of the size classes by Otoo et al. (2021), "the femur [of the class II] is similar to that of Class I but correspondingly larger in size (p. 731)", being seemingly incongruent with the relationship between FMNH PR 5022 and 5021. Some explanation is required regarding this problem in Results part (lines 115–118) and/or Discussion part (lines 154–157).”

We completely understand the apparent problem raised here and it is worth explicit clarification. In general, tetrapods, especially early ones, tend to have stouter limbs that become increasingly gracile and narrow during development even as their length increases. That is the case for *Whatcheeria* (see Supplementary Figure 1), and Otoo et al.'s (2021) comments about size increase refer to the length of the femur. During development, the shaft elongates and the articular surfaces become broader; as this happens the cortex thins such that a smaller percentage of the overall cross sectional area is made of cortex.

“Minor comments.

(2) Supplementary Fig. 1.

I recommend the authors to indicate the planes of cross sections as lines on the photos of the entire specimens. It will help readers to understand the positions of the cross sections.”

This is helpful comment and we have added plane of section guide for the reader. Thank you.

(3) Supplementary Table 1.

The caption should be placed above the table.

We have made this adjustment.

Reviewer 2:

“Dear Editor, Dear Authors,

Please find enclosed my review for the manuscript "Fossil bone histology reveals ancient origins for rapid juvenile growth in tetrapods" (COMMSBIO-22-2182A) submitted to Communications Biology.

This is the second round of reviews for this manuscript. The authors have done a great job incorporating the suggestions of both reviewers in this new version – uncertainties regarding ontogeny, intraspecific variability, and potential similarities in growth patterns with other tetrapods are now discussed with corresponding references, in a way that allows the reader to look for more information if necessary. I am also really happy with the added systematics subsection of the Methods, which clarifies terminological ambiguities regarding the affinities of *Whatcheeria* among tetrapods and close outgroups. Finally, the authors have carefully edited aspects of their work related to metabolic inference, which makes the discussion on growth patterns in the context of metabolic heat production much more cautious. Overall, I see no further issues with this version – this is an excellent paper that deserves to be accepted as such in *Communications Biology*. Congratulations to the authors, and thanks for taking the time to answer our comments and queries.”

We thank the reviewer for their time and support of our manuscript!